# Occupational justice and social inclusion among people living with HIV and people with mental illness: a scoping review

Clement Nhunzvi [1], Lisa Langhaug,[2] Edwin Mavindidze,[3] Richard Harding [4], Roshan Galvaan[5]

For numbered affiliations see end of article.

**Correspondence to**
Clement Nhunzvi;
clemynhu@gmail.com

## ABSTRACT

**Objective** To explore ways in which occupational justice and social inclusion are conceptualised, defined and operationalised in highly stigmatised and chronic conditions of mental illness and HIV.

**Design** This scoping review protocol followed Arksey and O'Malley's (2005) Scoping Review Framework.

**Data sources and eligibility** The following databases were searched for the period January 1997 to January 2019: Medline via PubMed, Scopus, Academic Search Premier, Cumulative Index to Nursing and Allied Health Literature (CINAHL), Africa-Wide Information, Humanities International Complete, Web of Science, PsychInfo, SocINDEX and grey literature.

Eligible articles were primary studies, reviews or theoretical papers which conceptualised, defined and/or operationalised social inclusion or occupational justice in mental illness or HIV.

**Study appraisal and synthesis** We undertook a three-part article screening process. Screening and data extraction were undertaken independently by two researchers. Arksey's framework and thematic analysis informed the collation and synthesis of included papers.

**Results** From 3352 records, we reviewed 139 full articles and retained 27 for this scoping review. Definitions of social inclusion and occupational justice in the domains of mental illness and HIV were heterogeneous and lacked definitional clarity. The two concepts were conceptualised as either processes or personal experiences, with key features of community participation, respect for human rights and establishment and maintenance of healthy relationships. Conceptual commonalities between social inclusion and occupational justice were premised on social justice.

**Conclusions** To address lack of clarity, we propose further and concurrent exploration of these concepts, specifically with reference to persons with comorbid mental health disorders such as substance use disorders and HIV living in low-income countries. This should reflect contextual realities influencing community participation, respect for human rights and meaningful occupational participation. From this broadened understanding, quantitative measures should be applied to improve the standardisation of measurements for occupational justice and social inclusion in policy, research and practice.

## Strengths and limitations of this study

► The methodology as provided by the scoping review design, facilitated comprehensive mapping of the literature, and presented a foundation for further exploration utilisation of the concepts to inform policy, research and practice.

► We used a rigorous strategy to explore research foci, definitions and utilisations of the concepts of social inclusion and occupational justice in mental illness and HIV.

► Data synthesis was limited to work published in English originally or with available English-translated copies.

► We focused on mental illness, which is made up of several different conditions and could have introduced generalisation bias. However, most of the included studies were also not condition specific, fitting our primary aim for conceptual review.

► Focus was limited to conceptual and theoretical aspects of the concepts more than interventions and outcomes of interventions.

## INTRODUCTION

The global burden of disease from HIV remains substantially high with approximately 37.9 million people living with HIV.[1] However, in the last two decades, the world has seen a combination of a significantly decreased mortality and a small decrease in incidence leading to an increase in the number of people living with HIV from 8·74 million (1990) to 36·9 million (2017).[2] This increase in the number of people living with HIV and decreased mortality rates are largely a result of the scaling up of HIV treatments.[1 3] However, what remains as a concern are persisting gaps in the treatment continuum towards the UNAIDS (The Joint United Nations Programme on HIV and AIDS) 90–90–90 target. Among those living with HIV who knew their status globally, 17% were still not on life-saving antiretroviral

therapy (ART) UNAIDS Global AIDS update 2019.[1] Moreover, only 53% of those on treatment were virally suppressed.[1 3] One reason for these discrepancies, mainly seen in key populations, is the rising and ever complex relationship between mental illness and HIV.[4 5] Secondary to both biological and psychosocial factors, people living with HIV are at an increased risk of experiencing poor mental health[6 7] which negatively impacts on their health-seeking behaviours, adherence to antiretroviral treatments[8] and quality of life.[9] The prevalence of common mental disorders is also significantly higher among people living with HIV, irrespective of them being on ART, and is further impounded by stigma.[10] There is also a known bidirectional relationship between HIV and mental health, worsened by associated health and social inequalities. This often leaves people with severe mental illnesses at an increased risk for HIV infection.[11 12]

Individuals with chronic and usually stigmatised conditions, such as mental illnesses, physical disabilities and HIV, face barriers to full participation in their communities.[13] Poverty, lack of access to education, lack of suitable housing and unemployment are some of the social and economic barriers to accessing adequate and sustained healthcare faced by this group.[14] These barriers may similarly be experienced by people who face discrimination based on their class, race or gender identity or sexuality and thus, when persons with mental illness also share these characteristics they may be severely stigmatised or discriminated against.[15 16]

To sustain the aforementioned progress in the management of people with HIV and its comorbidities such as mental illness, there is a renewed call to take a community-led, equality and social justice approach[3 12] with concepts like occupational justice and social inclusion holding promise to inform this agenda. Occupational justice is an advanced form of social justice, concerned with equity and fairness for individuals, groups and communities access to resources and opportunities that supports their engagement in diverse, healthy and meaningful occupations.[17 18] On the other hand, social inclusion entails multidimensional processes or states where prevailing contextual conditions enable full and active participation in all aspects of everyday life.[19 20] This can include civic, social, economic and political activities, as well as participation in decision-making processes irrespective of personal characteristic differences.[19 20]

Social inclusion and occupational justice form aspects of social justice and are therefore relevant to direct research and practice as we address exclusions and injustices experienced by stigmatised groups.[21]. Focus on these social justice outcomes ensures that health and social well-being are addressed beyond the clinical management of the disease. Health-related quality of life that includes social inclusion and occupational justice will be a holistic construct for promoting continuum of care and health and well-being beyond viral suppression in HIV.[22 23] These concepts direct the health and social care communities to view individuals with mental illness and HIV as being part of marginalised groups at risk of being deprived of respect, rights and opportunities to achieve optimal health-related quality of life.

Social inclusion and occupational justice are potentially key concepts that can inform the promotion of human rights-based, sustainable, person and community-centred interventions that promote recovery for persons with chronic and stigmatised conditions.[24 25] In order to aid integration and operationalisation of occupational justice and social inclusion in practice, we need to understand how the concepts are conceptualised and applied in population groups affected by chronic and stigmatised conditions. Synthesised summaries of research evidence can inform primary research and implementation science,[26] therefore we selected a scoping review design to help advance this field.[27] This was a particularly appropriate method for this area due to the diverse disciplinary locations of the existing literature.[28 29] This scoping review aimed to explore and appraise the definitions, current utilisation and relationships between the concepts of social inclusion and occupational justice in mental illness and HIV literature.

## METHODS

This scoping review followed our published study protocol,[27] developed using Arksey and O'Malley's Scoping Review framework,[29] as well as guidelines for scoping review protocols in occupational therapy.[26 30] In this paper, a scoping review is taken to be a form of knowledge synthesis that addresses an exploratory research question rather than the highly focused question in a systematic review.[28 29] For reporting, we followed the PRISMA extension for Scoping Reviews (PRISMA-ScR) Checklist.[31]

We followed an iterative process to develop and refine the research question.[29] Based on the subject area terminology, literature and our understanding of current practice trends in managing conditions that are chronic, and stigmatised, we asked the following question:

How are occupational justice and social inclusion conceptualised, defined and operationalised, and how are these concepts related in the highly stigmatised chronic conditions of mental illness and HIV?

The objectives of our scoping review were
1. To identify articles that define or conceptualise occupational justice and social inclusion related to mental illness and/or HIV.
2. To describe how these are operationalised or utilised.
3. To identify and describe relationships between occupational justice and social inclusion.
4. To determine potential areas for further development, integration, and application of these concepts.

## Search strategy

With the aid of a subject librarian, we identified appropriate databases using a journal indexing system. We

| Table 1 | General search strategy |
|---|---|
| **Key word** | **Alternative words** |
| Occupational therapy | Occupational rehabilitation |
| | AND |
| Mental illness | Mental health OR Mental disorder OR Psychiatric disability |
| | AND |
| Occupational justice | Occupational injustice OR Occupational marginalisation OR Occupational alienation OR Occupational imbalance OR Occupational deprivation |
| | AND |
| Social inclusion | Social exclusion OR Social isolation OR social integration |
| | AND |
| HIV | HIV OR HIV/AIDS OR HIV infection OR AIDS |

searched 12 databases in January 2019: (1) PubMed, (2) Scopus, (3) Academic Search Premier, (4) the Cumulative Index to Nursing and Allied Health Literature (CINAHL), (5) Africa-Wide Information, (6) Humanities International Complete, (7) Web of Science, (8) PsychInfo, (9) SocINDEX, (10) Grey Literature Report, (11) Web of Science Conference Proceedings and (12) Open Grey. We used PubMed as the free platform for accessing articles indexed on Medline database. The selected databases captured a comprehensive sample of literature from a variety of disciplines including social work, psychiatry, nursing and occupational therapy.

The first and last authors (CN and RG) worked with the librarian, through an iterative process, to develop an inclusive list of search terms and applicable filtering methods including Boolean phrases and Medical Subject Headings (MeSH) terms for each database.[27] We developed a general search strategy with primary search terms related to the primary concepts of occupational justice and social inclusion, while secondary search terms encompassed the broader terms of mental health, occupational therapy, mental illness, HIV and rehabilitation (table 1). For the purposes of search strategy development, we restricted our search to literature published between January 1997 and January 2019, a period which has seen the emergence and rapid growth in literature on occupational justice.[32]

We conducted a preliminary search on PubMed and this enabled refinement of our search strategy to maximise sensitivity and specificity. We adapted the PubMed search strategy (online supplementary appendix 1) accordingly for other databases.

### Study selection

The first author (CN) reviewed the titles identified in the search for eligibility. The aim was to identify articles that (1) indicated a research focus on mental illness, or HIV, or both and (2) titles that included the key terms of occupational justice, social inclusion or both. Article types included primary studies, reviews, opinion papers and other theoretical papers without primary data. Articles were not eliminated where there was uncertainty with the title until it was examined more in-depth by looking at the abstract. Two independent reviewers, the first and third authors (CN and EM), reviewed titles and abstracts of preliminarily selected articles using predetermined inclusion and exclusion criteria (table 2), detailed in our protocol.[27] These same reviewers each further screened full-text articles to determine if they met the inclusion/exclusion criteria. At this stage, articles were included if their explicit focus was on social inclusion or occupational justice in mental illness and/or HIV; concepts were defined and reported some operationalisation of the two concepts. Discrepancies were resolved by consensus or by seeking adjudication from the second author (LL). The Cohen's κ statistic to determine inter-rater agreement was calculated for the title and abstract review and the full article review stage, giving more than 90% agreement between reviewers with Cohen's κ statistic of 0.78 and 0.83 respectively.

### Data extraction

Guided by the study objectives, a common extraction table was designed, to extract standard bibliometric information study characteristic and main findings. The first five articles were reviewed by both the first and third authors, with the remaining articles divided between the two authors. We then checked for accuracy and completeness against each other's work. Discrepancies were resolved by revisiting the article, discussing and reaching consensus.

| Table 2 | Inclusion and exclusion criteria for acceptable articles | |
|---|---|---|
| **Criteria for inclusion** | | **Criteria for exclusion** |
| Minimum criteria required in the abstract:<br>► Explicit mention of mental illness, and/or HIV/AIDS.<br>► Explicit mention of either occupational justice and its varieties or social inclusion and its varieties. (varieties were informed by how articles with these terms were indexed in databases)<br>► Date range (January 1997 to January 2019).<br>► English language<br>► English translation of abstract and article available | | ► Used animal subjects. |

### Data synthesis

Data were synthesised descriptively to give a structured summary of the dataset and to capture the characteristics of the studies included and the definitional range of social inclusion and occupational justice. Study grouping followed the publication trends over time and study designs used. Descriptive statistics were calculated using Microsoft Excel version 16.0 for frequencies. We used deductive thematic analysis to organise the extracted definitions and related concepts for occupational justice and social inclusion.

### Patient and public involvement

Patient and public involvement representatives were not directly involved in the design of this scoping review protocol. However, experiences of the first author in working with adults afflicted with HIV and mental health disorders in Zimbabwe informed the need to explore issues faced by this population beyond biomedical care. We also built our research question from insights being generated in his doctoral studies exploring occupational perspectives on social inclusion among young adults dually afflicted with substance use disorders and HIV. Social inclusion speaks to life beyond medical management which was not being given sufficient attention and hence the need to conduct a scoping review.

## RESULTS

### Retained studies' characteristics

As described in figure 1, we screened titles (n=3352) and after reviewing (n=139) full articles 27 were included in this scoping review. Of the (n=27) sources included for final review, 23 were published between 2009 and 2018, with the majority (n=6) of these published in 2012 (table 3). Most publications were by authors in the mental health field and from high-income countries, with 68% of the primary studies being conducted in Europe,[33–45] and 9% in Australia.[46 47] No primary studies were conducted in North America, Africa or Asia. More than a third (n=10) of the studies utilised a qualitative research design,[34–37 40–42 44 46 48 49] five (16%) utilised a quantitative research design[33 38 39 45 47] and only one study utilised a mixed-methods design.[43] Six (19%) were review papers and the remainder (n=5) were opinion,[50 51] lecturership,[52] commentary[53] and theoretical analysis papers.[54] The two concepts were predominantly explored around mental illness (n=26) with significantly less focus on HIV (n=1). The majority (n=21) of the published research investigated social inclusion as it related to mental health conditions, and all the occupational justice papers were focused on mental illness. Only one paper was included which discussed social inclusion in relation to people living with HIV.[55] We found no published literature that explored occupational justice and social inclusion in populations with comorbid mental illness and HIV.

## SOCIAL INCLUSION: CONCEPTS AND DEFINITIONS

Social inclusion was defined with high variability, with only two studies using the same definition by defining social inclusion as a subjective sense of belonging and active citizenship that enhances social integration.[40 42] Conceptualisations and definitions used ambiguous words such as 'community', 'participation' and 'integration' to define social inclusion. Some studies defined social inclusion in terms of paid work and participation in community events,[38 39] others focused on social acceptance and absence of stigma,[40 42 45 56] while still others saw it as a political discourse.[41] Stain *et al*[47] tried to capture these varieties, and defined social inclusion as

> the participation of a person in society, evidenced by an individual having the opportunities, resources and abilities to build and maintain relationships, engage in education and employment, and participate in community events and organisations. (p880)

Notable thematic areas emerged from the analysis of the definitions, namely community participation, human rights and social relations that enhances a sense of acceptance and belonging.

### Community participation

The most prominent shared features of the definitions of social inclusion in mental illness and HIV research focused on it being a process and an experience centred on community participation. However, the terms 'community participation' and 'community' were used in many ways without clear descriptions. These varied interpretations of community participation included reference to people with HIV or mental illness were wide ranging and included individuals with opportunities to participate in key activities in their communities like paid employment[38 39 47 57]; being integrated into the community[37 41 45]; having a sense of belonging within the community[33 40 42 51]; and exercising active citizenship.[39 55 58]

### Human rights

Social inclusion was also defined and conceptualised as a human rights issue,[39 46 49 56 58] even though community participation was the penultimate indicator for social inclusion. When discussing human rights, the authors highlighted an individual's right to access resources and opportunities for personal and community growth. Social inclusion was also conceptualised as the right to engage in productive occupations, with full access to work and/or educational activities within the community despite one's health concerns.[58]

### Social relations

Social inclusion was further conceptualised and defined as a subjective experience,[33 40 42 43 45 47] where those who are socially included should experience positive relationships. While a number of authors[33 40 42 43 45 47] talked about relationships as being a key component, this was not defined or discussed in depth. Instead it was emphasised that for social inclusion to be a reality, an individual should

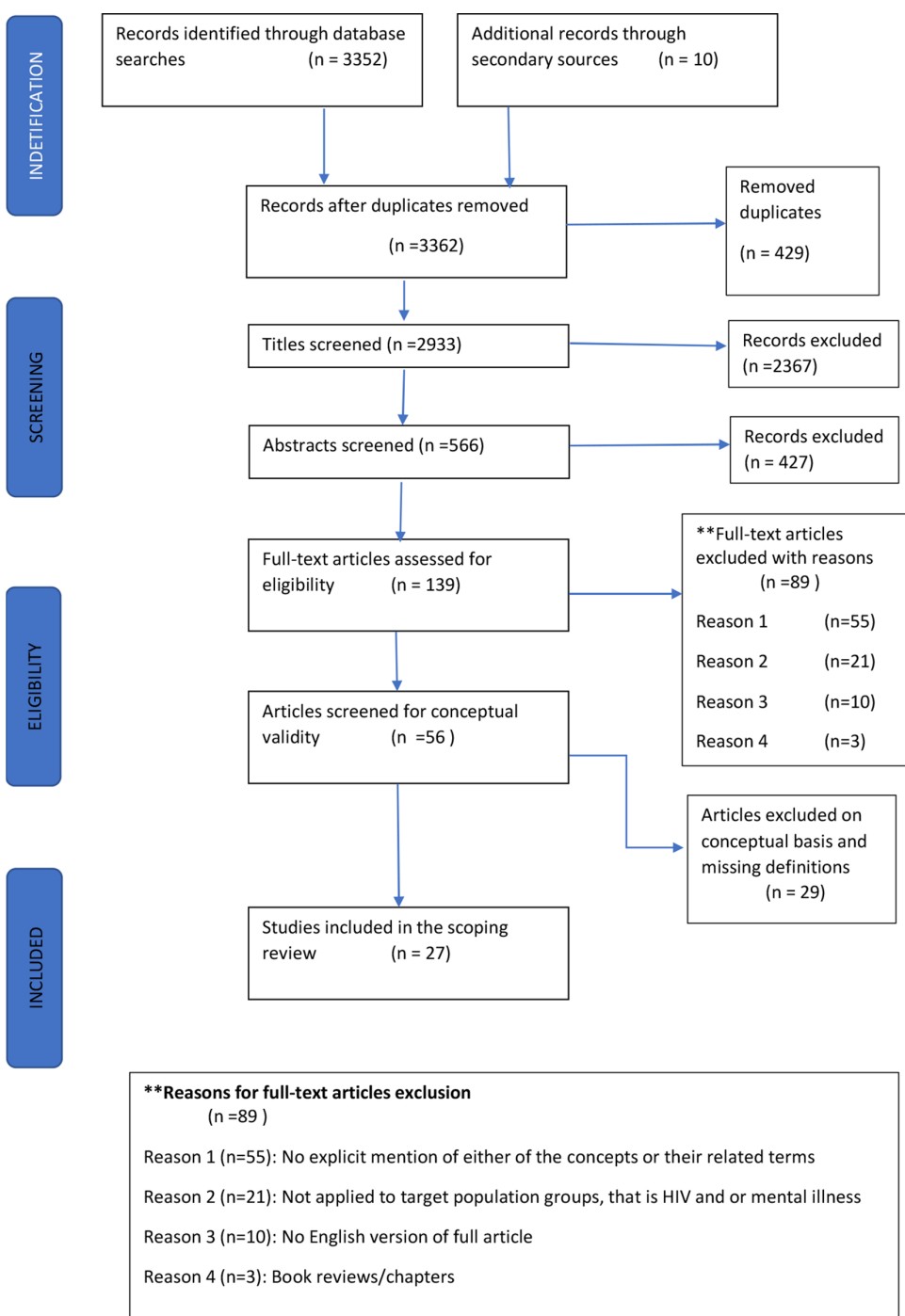

**Figure 1** Scoping review flow diagram.

experience positive social relationships with their significant others, family, friends and acquaintances.[33 40 42 43 45 47] Social inclusion was also conceptualised as experiencing social support and having positive support networks.[45]

### Diversity in definitions reflected in measurement tools

The diversity in definitions and features was also present in the measurement tool used. Quantitative studies included one of the following social inclusion measures: (1) Social Inclusion Questionnaire,[43] (2) Social Relationships Scale and Social Inclusion Scale,[33] (3) Social Inclusion Interview Schedule[45] and (4) Social Inclusion Questionnaire

User Experience.[38] Social inclusion was also portrayed in some definitions as a subjective personal concept, where it is the individual with mental illness or person living with HIV, who subjectively experiences inclusion and should have a choice on what determines their experience of such inclusion.[34 40 42 44 51] It is their perception of the quality of their relationships, their acceptability to others, and how integrated they are, which was emphasised.

In summary, social inclusion was conceptualised as processes and experiences of empowered and equitable community participation for all, in which there is respect

**Table 3** Summary of articles defining social inclusion and occupational justice in mental illness and HIV

| Author(s) year | Participants Setting | Design | Aim | Definition(s) of social inclusion and/or occupational justice | How operationalised | Associated key terms used |
|---|---|---|---|---|---|---|
| *3a: Studies with primary data collection* | | | | | | |
| Mazzi et al 2018[39] | 30 people with non-affective psychosis. In a social inclusion intervention programme<br><br>Italy | Cross-sectional study | 'To assess whether a social inclusion intervention is associated with better outcomes in terms of personal and social recovery, with particular reference to the areas of social functioning and activity, and subjective dimensions such as self-esteem, self-stigma and perceived quality of life' (p1) | Social inclusion is the opportunity for an individual to participate in key functions or activities and in the economic, social and cultural life of his/her community, exercising the rights of his/her citizenship and enjoying an adequate standard of living and well-being. | As an outcome and intervention<br><br>As a human rights issue | social withdrawal, quality of life, discrimination, social recovery, social inclusion activities, social disadvantage |
| Saavedra et al 2018[43] | 31 service users with severe mental illness. In workshops for social integration<br><br>Spain | Mixed-methods study | To evaluate the impact of an artistic workshop on a group of people diagnosed/ screened for with severe mental illness with focus on the impact of creative practices on well-being and social inclusion outcomes. | Social inclusion stated as a personal construct measured through perception of social isolation, social relationship and social acceptability | As an outcome to be evaluated based on personal perception<br><br>Measured using Social Inclusion Questionnaire | stigma, wellbeing, recovery, social isolation, social relationship, social acceptability |
| Berry and Greenwood 2017[33] | 51 young outpatient service users with first episode psychosis. In Early Intervention in Psychosis, Community Mental Health and Assertive Outreach services<br><br>UK | Longitudinal study | 'To investigate the direct and indirect associations between dysfunctional attitudes, self-stigma, hopefulness, social inclusion and vocational activity for young people with psychosis' (p197). | Social inclusion comprises social activity and community belonging. | As measured by levels of social activity and community belonging using Social Relationships Scale and Social Inclusion Scale | self-stigma, hopefulness, social activity, community belonging, social network, social contact, vocational activity |
| Turner et al 2017[45] | 71 people with psychotic-related conditions. In a research programme on schizophrenia<br><br>Ireland | Quantitative descriptive study | To explore 'the level of social inclusion among people with psychotic-related conditions using a standardised interview' (p195) | Social inclusion is a multidimensional phenomenon with a number of domains including socially valued role functioning, social support, absence of stigma and integration in rehabilitation community and wider community | As a multidimensional phenomenon<br><br>As both objective and subjective outcome<br><br>Assessed using Social Inclusion Interview Schedule | supportive relationships, stigma, integration, social exclusion, social support, rehabilitation |

**Table 3** Continued

| Author(s) year | Participants Setting | Design | Aim | Definition(s) of social inclusion and/or occupational justice | How operationalised | Associated key terms used |
|---|---|---|---|---|---|---|
| Raitakari et al 2016[41] | 16 mental health service users. In two mental health FSSs<br><br>Finland | Qualitative study | To explore how community integration is understood and tackled in mental health FSSs and, more precisely, in service user–practitioner home visit interaction | Social inclusion is articulated in political discourse as a way to tackle social exclusion and to increase citizens' participation and activity in society | As supported integration in community life<br>As participation in what marginalised individuals are usually excluded from | social exclusion, citizenship, community integration, housing, interaction, mental health, recovery |
| Killaspy et al 2014[38] | 67 mental health service users with psychosis. Living in the boroughs of London and the inner-city<br><br>UK | Quantitative study | To investigate change in social inclusion after the development of a psychotic illness (clinically diagnosed) and associated factors | 'Social inclusion refers to the opportunities that individuals have to participate in key areas of economic, social and cultural life' (p148) | Focus on participation and access to services and opportunities<br><br>Five domains of social inclusion: social integration, consumption, access to services, productivity, political engagement<br>Assessed using the Social Inclusion Questionnaire User Experience | social exclusion, social disadvantage, social integration, employment, social isolation |
| Salles and Barros 2013[49] | 17 mental health service users and 12 individuals from their social networks. In Psychosocial Care Centres<br><br>Brazil | Qualitative study | 'To identify and analyse the conceptions expressed by the interviewees about social inclusion and mental illness' (p37) | 'Social inclusion is a process of promoting rights, access, choice and participation. For individuals with mental health problems, this also means access to the best possible forms of treatment' (p37) | As a dynamic, multidimensional process | work participation poverty |
| Salles and Barros 2013[49] | 17 mental health service users and 12 individuals from their social networks. In Psychosocial Care Centres<br><br>Brazil | Qualitative study | 'To identify the daily life experiences of users of a psychosocial care centre related to processes of social exclusion and inclusion' (p704). | 'Social inclusion is a process of the individual's choice on how to live her daily life' (p710) and not just about experiencing full participation in society | As a multidimensional process<br>As a human rights issue<br>As subjective experience centred on work participation | social exclusion, discrimination, prejudice, stigma, social isolation, social participation, empowerment, poverty |

Continued

**Table 3** Continued

| Author(s) year | Participants Setting | Design | Aim | Definition(s) of social inclusion and/or occupational justice | How operationalised | Associated key terms used |
|---|---|---|---|---|---|---|
| Clewes et al 2013[34] | Single case-study with bipolar depression. In occupational therapy interventions under UK National Health Services (NHS)<br><br>UK | Case study | To illustrate how the combination of medical outpatient clinic and occupational therapy intervention together made a big difference in a person's life. | Social inclusion was taken as meaningful inclusion in the areas of life where the participant wished to be. | As personal experience<br><br>As a degree of autonomy, ownership, responsibility and independence | stigma, recovery, social policy, empowerment, rights, engagement, spirituality, client leading |
| Nieminen et al 2012[40] | 23 mental health service users. In an intervention group follow-up study<br><br>The European Union | Qualitative study | To describe how the mental health service users experienced social inclusion and employment in the European Union EMILIA project | Social inclusion is a subjective sense of belonging and active citizenship that enhances social integration | As an experience and feeling of active citizenship | empowerment, social network, finance and housing employment, social exclusion, stigma, prejudice |
| Fieldhouse 2012[36] | Eight mental health service users in a 2 year action research project<br><br>UK | Qualitative research | To describe an action research project that explored the recovery journeys of a group of assertive outreach service users who had progressed from being socially excluded and occupationally deprived to being participants in their local communities and to use this knowledge to inform local service development. | Social inclusion is when people with mental health challenges enjoy rewarding social relations through renewed engagement in mainstream occupations resulting in fuller community participation | As active participation in context or local community<br><br>As an outcome of occupation-based interventions | social participation, social relationships, stigma, social exclusion, community participation, belonging, social capital |
| Stain et al 2012[47] | 1825 adults with psychosis in an Australian national survey of psychosis<br><br>Australia | Quantitative study | 'To explore the impact of psychosis on an individual's social and community participation' (p879) | 'Social inclusion refers to the participation of a person in society and is evidenced by an individual having the opportunities, resources and abilities to build and maintain relationships, engage in education and employment and participate in community events and organisations' (p880) | As participation in mainstream life as empowered individuals | social isolation, social anxiety, stigma, social participation, community participation |

Continued

**Table 3** Continued

| Author(s) year | Participants Setting | Design | Aim | Definition(s) of social inclusion and/or occupational justice | How operationalised | Associated key terms used |
|---|---|---|---|---|---|---|
| Smyth et al 2011[44] | Eight mental health service users in mental health rehabilitation services in inner-city area<br><br>UK | Qualitative study | To explore the experiences of social inclusion for mental health service users and factors associated when engaging in everyday community occupations | Social inclusion is developing fair access to opportunity in key social and economic spheres for marginalised groups | As an issue of participation and access to services and opportunities | stigma occupational deprivation, discrimination, social networks |
| Ramon et al 2011[42] | 27 key informants in an evaluation study of the EMILIA project<br><br>The European Union | Qualitative study | To identify how participation in the EMILIA project affects the lives of mental health service users in relation to social inclusion, employment and recovery | Social inclusion is a subjective sense of belonging and active citizenship that enhances social integration | As an experience and feeling about participation in society | employment, well-being, quality of life, social networks |
| Hamer et al 2017[46] | 82 mental health service users<br><br>New Zealand and Brazil | Qualitative study | To present service users' stories of distressing exclusion that interrupted their rights to occupational justice, and marginalised them from occupation. The paper also presents the practices of inclusion that service users engaged in that restored their rights and responsibilities as occupied and active citizens | Social inclusion is the extent to which people are confident about and able to exercise their rights and participate, by choice, in the ordinary activities of citizens<br><br>Occupational justice recognises the person's right to inclusive participation in everyday occupations regardless of age, ability, gender, social class or other differences | Social inclusion as a policy issue<br>As a dynamic concept, derived from subjective experiences<br><br>Social inclusion is enhanced through occupational justice<br><br>As inclusive participation<br>As a human right issue | occupational injustice, citizenship, stigma, discrimination |
| Fieldhouse 2012[36] | Eight mental health service users in a 2 year action research project<br><br>UK | Qualitative Study | To examine the impact of community participation on their recovery and social inclusion and how service users' experiences informed joint planning between mental health services and the learning community to promote social inclusion. | Social inclusion as the process of enabling citizenship through fuller community participation Occupational justice defined as the process of lobbying for the occupational needs of individuals and communities as part of a fair, inclusive, and empowering society; as a community reintegration issue | As active participation in mainstream society | social participation, social relationships, stigma, social exclusion, community participation, belonging |

Continued

**Table 3** Continued

| Author(s) year | Participants Setting | Design | Aim | Definition(s) of social inclusion and/or occupational justice | How operationalised | Associated key terms used |
|---|---|---|---|---|---|---|
| Farrell and Bryant 2009[35] | Nine recruiters of volunteers who had mental illness UK | Qualitative study | To explore the recruiters' understanding of mental health problems, drawing on their experiences | Occupational justice is an intrinsic part of social justice, permitting equitable opportunity and the means to choose, organise and perform meaningful occupations  Social inclusion stated but not defined | As a process and an outcome  Occupational justice as an intrinsic aspect of social inclusion | volunteering, discrimination, stigmatisation, social exclusion, social attitudes, prejudice, occupational deprivation, occupational marginalisation, occupational apartheid |
| *3b: Review, commentary, lectureship and opinion papers* | | | | | | |
| Le Boutillier and Croucher 2010[51] | Mental health service users | Opinion paper | To present an alternative to the polarised view of social inclusion | Social inclusion is a multidimensional virtuous circle aimed at improving rights of access to the social and economic world, new opportunities, recovery of social identity and meaningful life and also reduced impact of disability on everyday life | As a multidimensional system | social exclusion, occupational justice, occupational balance, occupational alienation, occupational deprivation |
| Cobigo and Stuart[56] 2010 | Mental health service users | Review | To review recent research on approaches to improving social inclusion for people with mental disabilities | Social inclusion is when one feels accepted and recognised as an individual beyond the disability; has positive personal relationships with family, friends and acquaintances; is actively involved in recreation, leisure, and other social activities; has appropriate living accommodations; has healthy employment; and has appropriate formal (service system) and informal (family and caregiver) supports | As an acceptance, human rights, outcome of interventions issue | stigma, discrimination, legislation, community support, disability rights, justice, human rights |

Continued

**Table 3** Continued

| Author(s) year | Participants Setting | Design | Aim | Definition(s) of social inclusion and/or occupational justice | How operationalised | Associated key terms used |
|---|---|---|---|---|---|---|
| Cáceres et al 2008[55] | Global literature on Men who have sex with men (MSM) with HIV | Review | To analyse reasons for continued risk of HIV and its consequences in MSM globally | 'A social inclusion perspective on HIV prevention and AIDS care implies the adoption of a broad range of strategies to understand and confront social vulnerability' (p11) | As a perspective to addressing vulnerability | social exclusion—describes the alienation or disenfranchisement that certain individuals or groups experience within society—stigma discrimination, prejudice, human rights, poverty, migration, employment participation, sexuality |
| Lloyd et al 2006[66] | Mental health service users | Review | To describe a selected number of activities that promote social inclusion | Social inclusion involves being able to rejoin or participate in leisure, friendship and work communities | Participating and accessing services and opportunities | connectedness and interdependence |
| Farone 2006[70] | Mental health service users, with a focus on schizophrenia | Review | To examine empirical evidence describing experiences with social or community integration for people with psychiatric disabilities, with a particular interest in schizophrenia | Social inclusion discussed but not defined | A link to mental and emotional well-being | community integration, community inclusion, social integration, stigma, social support, social networks |
| Evans and Repper 2000[57] | Mental health service users | Review | To challenge common misconceptions surrounding employment, work and mental health problems of mental health service users | Social inclusion is defined as a need, aspiration and citizenship issue among mental health service users Social inclusion as an outcome of work participation. | A social need An outcome of work participation | social exclusion, stigma, unemployment, poverty |
| Mandiberg 2012[53] | People with psychiatric disabilities | Commentary | To describe the failure of social inclusion as a concept and present an alternative approach through community development | Social inclusion refers to full participation in the broader community for people with severe mental illnesses | As experience of participation in the broader community | work integration, community development, social enterprises |

Continued

**Table 3** Continued

| Author(s) year | Participants Setting | Design | Aim | Definition(s) of social inclusion and/or occupational justice | How operationalised | Associated key terms used |
|---|---|---|---|---|---|---|
| Townsend 2012[52] | Mental health service users<br><br>Canada | Lectureship | To propose for an interdisciplinary knowledge exchange with a critical occupational perspective on the question: What lessons on boundaries and bridges to adult mental health can be drawn by connecting the capabilities and occupational frameworks of justice? | Occupational justice as the enjoyment of the 'occupational rights' of all people to engage and be socially included in their desired occupations, and thereby to contribute positively to their own well-being and the well-being of their communities | As a human rights, capabilities and justice issue | occupational rights, occupational possibilities, occupational deprivation, alienation, imbalance and marginalisation |
| Harrison and Sellers 2008[50] | Mental health service users and mental health team<br><br>UK | Opinion Paper | To explore the implications and challenges for occupational therapy roles in mental health services regarding socially inclusive practice and to discuss policy that is designed to broaden professional roles | Occupational justice identifies inequalities in opportunities to participate in occupations | As a human rights issue about participation in occupations<br><br>As a policy issue | social exclusion, participation, poverty, occupational deprivation |
| Hamer 2017[58] | Mental health service users | Review | To discuss how social inclusion for mental health service users can be enhanced through occupational justice and the protection of their rights as citizens to have meaningful employment. | Social inclusion can be defined as the extent to which people are confident about and able to exercise their rights and participate, by choice, in the ordinary activities of citizens Occupational justice stated but not defined | As a human right issue | social exclusion, stigma, occupational justice, employment participation, poverty |

Continued

**Table 3** Continued

| Author(s) year | Participants Setting | Design | Aim | Definition(s) of social inclusion and/or occupational justice | How operationalised | Associated key terms used |
|---|---|---|---|---|---|---|
| Pettican and Bryant 2007[54] | Mental health service users | Theoretical analysis | To explore the potential of occupational justice and its related concepts. To provide the occupational therapy profession with a theoretical justification for occupational therapists adopting an occupation-focused role in community mental health teams | Social inclusion is a drive aiming to overcome discrimination and stigma faced by people with mental health problems, in order to facilitate their having equal access to mainstream employment, education and leisure opportunities.

Occupational justice is recognising and providing for the occupational needs of individuals and communities as part of a fair and empowering society | Social inclusion conceptualised as a policy drive issue

As a justice and human rights issue
As an occupational participation issue | occupational deprivation, occupational imbalance, occupational alienation, social justice |

FSSs, floating support services.

for human rights and healthy social relations and well-being are promoted.

## CONCEPTUALISING AND DEFINING OCCUPATIONAL JUSTICE

In the last two decades, only six research papers on mental illness used the concept of occupational justice and provided a definition[35 36 46 50 52 54] (table 3). Five different definitions were found (table 3). As with social inclusion, occupational justice was defined with great variability, with it being referred to as both a process and as an experience.[35 36 46 50 52 54] Two major themes emerged pointing to social justice in which occupational justice was framed as an occupational rights issue and as a matter of community participation.

### Occupational rights

A key concept of occupational justice for individuals with mental illness was experiencing or enjoying one's occupational rights.[35 36 46 50 52 54] Occupational rights were taken to mean an individual's rights to participate in a range of meaningful and contextual occupations enabling them to flourish, fulfil their potential and experience life satisfaction in ways consistent with their contexts. Townsend defined occupational justice as the enjoyment of 'occupational rights' by all people to engage in occupations and feel socially included in their desired occupations, thereby contributing positively to their own well-being and to that of their communities.[52] Occupational justice was also highlighted as an advocacy process. where individuals could lobby for the occupational rights and needs of individuals and communities as part of an equitable, inclusive and empowering society.[36]

### Community participation

The process of promoting occupational justice was viewed as related to promoting social inclusion and community participation through advancing participation.[46] In another paper community participation was emphasised, with occupational justice defined as situations when people are seen as having the opportunity to choose to participate in the community.[35] The core emphasis in the definitions was the acknowledgement of occupational justice as a means of actioning community participation and social justice.[35 46]

However, in all the definitions there was an overuse of the term 'occupation' which forms part of the term we are working to define. While 'occupation' was framed as purposeful everyday activities people engage in, it carries diverse meanings outside occupational science and occupational therapy. This rendered many of the definitions and arguments cyclical in nature.

## COMMONALITIES BETWEEN SOCIAL INCLUSION AND OCCUPATIONAL JUSTICE

Although not clearly articulated, all the papers which discussed the two concepts together seemed to infer that occupational justice was an important ingredient of social inclusion.[35 36 46 52] Notwithstanding the considerable variability in definitions and conceptualisations of the two concepts, some commonalities were identified (figure 2). Both social inclusion and occupational justice seek to promote equitable access to opportunities for engagement and to seek for fairness and justice in an individual's community participation.[35 36 46 52]

Hamer *et al* provided the closest account of the relationship between the concepts positing that social inclusion

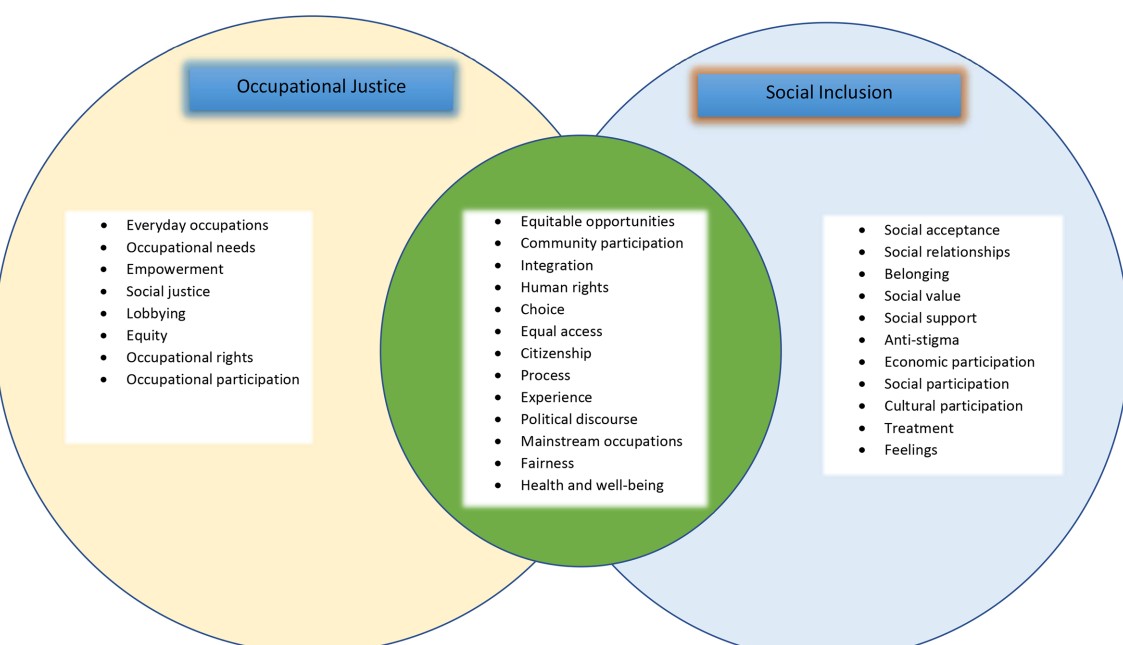

**Figure 2** Commonalities between occupational justice and social inclusion definitions—associated terms.

was enhanced through occupational justice.[46] Here they argued that recognition of people's right to inclusive participation in everyday occupations (occupational justice) enhanced the extent to which the person became confident about and was able to exercise their rights and participate by choice in the ordinary activities as citizens (social inclusion).[46] Both concepts were centred on key thematic areas of human rights, equality, inclusivity, and community participation.

Significantly, both social inclusion and occupational justice had a social justice focus, emphasising the right to inclusive participation in a community and individuals exercising choice of participation as part of their citizenry beyond their health conditions. Occupational justice specifies that the participation in meaningful occupations is central, while social inclusion highlights community participation. Both concepts also highlight the need to address discriminatory practices, by doing away with stigma.[45 46] Hamer and colleagues highlight that social inclusion is the process of experiencing inclusive participation in the community as a citizen, while occupational justice promotes social inclusion through participation in meaningful and valued activities.[46]

## DISCUSSION

We explored and appraised the definitions, current utilisation and relationships between social inclusion and occupational justice in the literature on mental illness and/or HIV. We identified and critically appraised 27 articles that presented a variety of definitions. Although we primarily sought out to describe the conceptualisations and utilisations of these concepts in the duality of mental illness and HIV, we found out that literature is scant and focuses on a single diagnosis. Most studies were on mental illness and conducted in high-income countries. Despite the great variability, key thematic concepts used to define social inclusion and occupational justice included community participation, human rights and relationships. The two concepts are theoretically related through a social justice focus, putting the emphasis on treating people with HIV and those with mental illness in a more respectful and equitable manner. The highlighted thematic concepts are central in directing research and practice toward the moral imperative of addressing exclusions and injustices experienced by people living with HIV, those with mental illnesses and other stigmatised groups.[21] There is also emphasis on the multidimensional nature of the concepts framed as both a process and a personal experience, also allowing a broader horizon of their application, from policy to practice. The main source of these definitions were qualitative studies using service user's experiences and experts opinions.

While definitions of social inclusion vary, our scoping review findings confirm that the definitions are still in line with the concept of poverty reduction and a focus on reducing stigma and discrimination.[44 45 55] Social inclusion emerged from European societies, in response to a welfare crisis and desire to fight disadvantage.[59] We propose that

this focus on welfare and fighting disadvantage should remain and should be taken up in the mental health and HIV fields, with poverty reduction, justice and equality as pillars of social inclusion. This would be even more effective if poverty reduction was prioritised as a specific focus of social inclusion when informing mental health and HIV policies, particularly in sub-Saharan Africa, where these problems are more prevalent and driven by poverty.[3]

The most striking observation was the lack of clarity in the definition of social inclusion, which is still evident, despite its existence in the literature for almost five decades.[59] Efforts to be all inclusive, multidimensional, person-centred and contextual can explain the variations and, thus, ambiguity in the definitions.[59 60] However, this lack of definitional clarity could hamper its universal concept utilisation, measurement and further exploration with a common goal.[61 62] This lack of a single, universal understanding has positive and negative implications for research and application of the concept in clinical practice. Without a universally agreed on definition, comparisons between studies and practice remains difficult.[62] This was echoed in the variety of social inclusion measures applied in the quantitative studies.[33 38 43 45] Given the variation in measures, a common understanding of what constitutes social inclusion in mental health has not been developed, despite calls for this in recent literature.[61–63] Therefore construct validity of measures is difficult to test. For the meantime, measures with a broader scope and cross-cultural validation like the Social and Community Opportunities Profile,[64 65] can be instrumental in developing standardised measures.

In contrast, the variations in defining the concepts reflect diversity and the importance of contexts, rather than a singular adoption of a 'universal' idea. Given the diversities in experiences of mental illness and living with HIV, where different regions have their own social, economic, political, cultural and historical realities influencing mental illness and HIV, conceptualisations of social inclusion may benefit from remaining open to multiple definitions in order to reflect the realities of different regions. It is possible to have multiple, but mutual interpretations that could be understood as complementary or even contesting, in line with contextual realities.

Embracing plural definitions may be especially important in low-income and middle-income countries, where such research needs further growth.[59] For example different activities purported to enhance social inclusion of people with mental health challenges, like paid employment, have different meanings and impact across regions and would influence how it is conceptualised as part of social inclusion. We found a preference towards paid employment as a key determinant of community participation in social inclusion among people with mental health challenges.[40 47 49 57 66] This can show the dominant knowledge systems of capitalism in societies,[59] where most of the studies were conducted. Other communal ways of engagement outside of a neo-liberal market structure could be explored as they may facilitate

new ways of understanding inclusion in the context of HIV and mental illness. In regions where HIV and mental illness are prevalent and complex health and social care problems, we advocate for exploration of social inclusion to generate contextual knowledge, that would inform socially inclusive policies, practice and further research.

Some of the social inclusion definitions had an individual focus, for example the individual had to be actively participating at the expense of the collective found in communal societies.[38 39 43] Given that the studies were mainly from high-income regions, there remains some missing voices in informing the definitions, given the largely communal orientation found in indigenous communities in low-income and middle-income settings. The challenge in some of the reviewed definitions was to try to focus on the individual and the impact of HIV and/or mental illness, yet the social justice agenda may better be approached from a population level with a focus on broader social determinants of health which has been the case in many countries addressing developmental and intellectual disabilities.[67] Defining social inclusion of people with mental ill health and HIV also needs to be done from the perspective of people who are in low-income countries, experiencing poverty, unemployment, social inequality and forms of violence, since people with these conditions are usually stigmatised. Discrimination and consequent social problems usually affect people as part of a collective rather than only as individuals. Hence a collective perspective that considers how groups of people are affected can strengthen the concept of social inclusion and promote its possible utility in low-resource practice contexts.

Though the concept of occupational justice has been present in the literature for about three decades,[32] we found limited evidence of its conceptualisation and application in mental health and no studies in HIV.[35 36 46 50 52 54] Despite the global justice theoretical orientation of the concept, the studies found were also exclusively from high-income countries,.[35 36 46] We found key features that could guide occupational justice utilisation and further theorisation in HIV and mental health: community participation by having one's occupational rights upheld, occupational needs met, empowerment and equity in occupational participation. To some extent, the lack of diversity in regions informing the concept offers an opportunity to strengthen it by adding insights from regional contexts with potential for different realities, experiences and viewpoints, such as Africa where HIV and mental illness are prevalent and intertwined.[68] Also some authors conceptualised occupational justice as participation in occupation(s) in an equitable manner,[46] they took participation as synonymous with justice, without spelling out the nature of the occupation and position of the person accessing the occupations. These have great potential in influencing how the accessed occupation impacts health, well-being and feeling of social inclusion among those with HIV and mental illnesses.

Despite the definitional lack of clarity, social inclusion and occupational justice are related concepts that can be used together to frame research and practice and inform policy in HIV and mental health. The commonality between the concepts is the need to promote equitable access to opportunities for community participation with fairness and equity for people with HIV and those suffering from mental illness.[35 36 46 52] The relationship between the concepts could be further developed using diverse communities to build evidence on how engagement in meaningful everyday activities underlie inclusive communities for people with mental illness and those living with HIV. This focus on occupational justice perspective presents an opportunity to routinely explore the nuances of everyday occupational participation and what that may mean for the process and experiences of social inclusion of those involved.

We therefore propose an expansion of the relationship between the conceptualisation of the two concepts, using most affected population groups, such as people with comorbid mental health disorders like substance use disorders and HIV in low-income contexts. This population group is known to have unmet broader health and social care needs hinged to the double stigma associated with substance abuse and HIV.[69] That expansion should reflect the contextual realities influencing community participation, respect of human rights and having healthy relationships, actioned through engagement in meaningful occupations. These contextually refined concepts of social inclusion and occupational justice should then be used together to inform policy, research and practice, for a just and inclusive society for those with stigmatised conditions like HIV and mental illnesses. The occupational justice and socially inclusive approach from policy through to practice will ensure health and social well-being outcomes are addressed beyond the medical management of mental illness and/or HIV. Health-related quality of life as a holistic construct for promoting continuum of care and health and well-being beyond viral suppression in HIV[22 23] will also be made practical with a social justice lens.

## CONCLUSION

To our knowledge, this scoping review is the first to appraise the concepts of occupational justice and social inclusion in populations afflicted by mental illness and HIV. Our findings have the potential to initiate critical conversations in the field and expand understanding and utilisation of occupational justice and social inclusion to critique and enhance global mental health. We have also presented commonalities which will give us a better theoretical foundation to inform further research, practice and training, especially from under-represented societies.

**Author affiliations**
[1]College of Health Sciences, Rehabilitation Department, University of Zimbabwe, Harare, Zimbabwe
[2]Department of Psychiatry, College of Health Sciences, University of Zimbabwe, African Mental Health Research Initiative (AMARI), Harare, Zimbabwe
[3]Occupational Therapy, Ingutsheni Central Hospital, Bulawayo, Zimbabwe
[4]Florence Nightingale Faculty of Nursing Midwifery and Palliative Care, Cicely Saunders Institute, King's College London, London, UK

[5]Health and Rehabilitation Sciences, University of Cape Town, Cape Town, South Africa

**Acknowledgements** The authors would like to thank Mary Shelton, UCT librarian for assistance in developing our search strategy. This article also benefited from writing soft skills and reviews by Dr Helen Jack through African Mental Health Research Initiative (AMARI).

**Contributors** All authors have made substantive intellectual contributions to the conduct and write up of this review. CN and RG conceptualised the review approach and provided general guidance to the research team. CN and EM were involved in systematic and independent screening and data extraction. CN provided primary input at all stages, developed all draft documents and had overall responsibility for the review. LL, RG and RH gave substantial review and critique through the review process and manuscript. All the authors reviewed and commented on the drafts of the manuscript and they all read and approved the final manuscript.

**Funding** This work was supported through the DELTAS Africa Initiative [DEL-15-01]. The DELTAS Africa Initiative is an independent funding scheme of the African Academy of Sciences (AAS) Alliance for Accelerating Excellence in Science in Africa (AESA) and supported by the New Partnership for Africa's Development Planning and Coordinating Agency (NEPAD Agency) with funding from the Wellcome Trust [DEL-15-01] and the UK government.

**Disclaimer** The views expressed in this publication are those of the author(s) and not necessarily those of AAS, NEPAD Agency, Wellcome Trust or the UK government.

**Competing interests** None.

**Patient consent for publication** Not required.

**Provenance and peer review** Not commissioned; externally peer reviewed.

**Data availability statement** No additional data available.

**ORCID iDs**
Clement Nhunzvi http://orcid.org/0000-0001-5804-9817
Richard Harding http://orcid.org/0000-0001-9653-8689

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
