## [Reviewer comments · BMJ Open]

ARTICLE DETAILS

TITLE (PROVISIONAL)	Occupational justice and social inclusion amongst people living with HIV and people with mental illness: A scoping review
AUTHORS	NHUNZVI, CLEMENT; Langhaug, Lisa; Mavindizze, Edwin; Harding, Richard; Galvaan, Roshan

VERSION 1 – REVIEW

REVIEWER	Angelo Brandelli Costa PUCRS - BRAZIL
REVIEW RETURNED	27-Feb-2020

GENERAL COMMENTS	This is an interesting article that presents a relevant contribution from a conceptual point of view to the field of HIV / AIDS. I suggest minor modifications. In terms of strengths and weaknesses, I suggest adding information regarding the study's conclusions. In the introduction, a broad definition of the investigated outcomes in the review is needed. In addition, there is a need for a more emphatic defense of the plurality in its definition that justifies carrying out the review.
---

REVIEWER	David Chipanta PHD Candidate, Biomedicine University of Geneva, UNAIDS, Geneva
REVIEW RETURNED	18-Mar-2020

GENERAL COMMENTS	The paper is interesting. The premise that social inclusion and occupational justice are key to not leave people with HIV and mental health behind is important. However the paper requires revision to provide clarity on a number of areas: Use UNAIDS sources in the introduction, and highlight the contribution of people accessing HIV treatment to the increasing number of people living with HIV. The authors need provide broad guidelines on what they understand social inclusion, social justice and occupational justice is how relevant these concepts are to the response to HIV and mental health. The link between HIV and mental health is casually touched on, and leaves the readers with a sense that people with HIV have mental
---

	illnesses. A bit more articulation of the relationship between mental health and HIV would be helpful .e.g. that not all people with HIV have mental illnesses and that not all people with mental illness have HIV. It is a useful paper and with some revision it add to the literature. The search strategy, search terms (.e.g. paper including "social inclusion, mental health and HIV") and inclusion criteria need to be clarified. The discussion does not say anything about people living with HIV and or mental illnesses and how the paper contributes to understanding and policies to include people living with HIV and mental illnesses.
--	--

VERSION 1 – AUTHOR RESPONSE

Reviewer: 1	In terms of strengths and weaknesses, I suggest adding information regarding the study's conclusions.	Thank you for this suggestion, we have added the point about the direction being set by this review.	Page 3
	In the introduction, a broad definition of the investigated outcomes in the review is needed. In addition, there is a need for a more emphatic defense of the plurality in its definition that justifies carrying out the review.	Thank you for these comments. This is a welcome suggestion which we have actioned by including the definitions and making a link to the justified call for this current review.	Introduction. Page 4-5
Reviewer: 2	Use UNAIDS sources in the introduction, and highlight the contribution of people accessing HIV treatment to the increasing number of people living with HIV.	Thank you for this review comment which we agree strengthens the background to this study. We have now included UNAIDS sources in supporting our assertions.	Introduction. Page 4
	The authors need to provide broad guidelines on what they understand by social inclusion, social justice, and occupational justice in how relevant these concepts are to the response to HIV and mental health.	This is a welcome suggestion which we have followed by including the broad definitions and further stated the justification for their importance as part of a holistic response to HIV and mental illnesses.	Introduction. Page 4-5
	The link between HIV and mental health is casually touched on and leaves the readers with a sense that people with HIV have mental illnesses. A bit more articulation of the relationship between mental health and HIV would be helpful.e.g. that not all people with HIV have mental illnesses and that not all people with mental illness have HIV.	Thank you for drawing our attention to this. We have attended to the suggestion and clarified the relationship between mental health and HIV with supporting literature.	Introduction. Page 4
	It is a useful paper and with some	We have attended to this	Search strategy

	revision it adds to the literature. The search strategy, search terms (.e.g. paper including "social inclusion, mental health and HIV") and inclusion criteria need to be clarified.	review comment by adding tables 1 and 2 showing the general search terms and inclusion criteria respectively. We also made reference to the attached appendix showing our search strategy on PubMed. We are also pleased to highlight that the search strategy, search terms and inclusion criteria are explained in more detail in the scoping review protocol we published earlier. Reference is now made to this protocol in the manuscript.	and study selection. Page 7-8
	The discussion does not say anything about people living with HIV and or mental illnesses and how the paper contributes to understanding and policies to include people living with HIV and mental illnesses.	Thank you for bringing this up. We had primarily focused on conceptual elements. We have since revised this section to bring to the fore the application elements in the context of HIV and mental illnesses, including proposals from our findings for policy direction, especially in settings where the problems of HIV and mental illness are prevalent and complicated by injustices.	Discussion. Page 26-29

VERSION 2 – REVIEW

REVIEWER	Angelo Brandelli Costa Brazil
REVIEW RETURNED	16-Apr-2020

GENERAL COMMENTS	I believe the article can be published in the present form.
---

REVIEWER	David Chipanta PHD Candidate, University of Geneva, Switzerland Senior Advisor Social Protection, UNAIDS, Geneva, Switzerland
REVIEW RETURNED	02-May-2020

GENERAL COMMENTS	The search strategy is not clear and specific enough. The search terms need to include a combination of "social inclusion" + "occupation justice" + "mental illness" + "HIV." e.g. "social inclusion" + "occupation justice" + "mental illness" + "HIV." "social inclusion" + "mental illness" + "HIV" ; "occupation justice" + "mental illness" + "HIV" or different variety and combination of the above. The discussions of community participation, occupation rights and
--

	social justice at the best only mention mental illness and not HIV. The utility of discussing HIV and or mental illness remains unanswered and confusing, since it is not clear if the discussion is on people with disability, people living with HIV or people living with HIV who may have mental illness. The paper will be much stronger with sharper focus and connections between community participation, occupation rights and social justice and mental illness or HIV. If the paper discuss to discuss both mental health and HIV, it needs to state so and highlight what aspects of mental health and HIV it will focus on. This is an important topic and with revisions it is upto important information.
--	--

VERSION 2 – AUTHOR RESPONSE

Reviewer	Original Manuscript and Reviewer Comments	Revised Manuscript and Author(s) Responses	Changes done on page/section
Reviewer: 1	I believe the article can be published in the present form.	Thank you for the review and recommendations which have seen the quality of our manuscript being improved.	
Reviewer: 2	The search strategy is not clear and specific enough. The search terms need to include a combination of "social inclusion" + "occupation justice" + "mental illness" + "HIV." e.g. "social inclusion" + "occupation justice" + "mental illness" + "HIV." "social inclusion" + "mental illness" + "HIV" ; "occupation justice" + "mental illness" + "HIV" or different variety and combination of the above.	We thank the reviewer for this important observation, detail of the search strategy is a key component of a review. We have included the search terms within the full search strategy description and report. We provided this as Appendix 1 (PubMed Search Strategy) We then adapted this to suit the search platforms for other databases. Below is the detailed search strategy we used: PubMed Set 1 1. Social Justice [MeSH] 2. Social Isolation [MeSH] 3. Social Marginalization [MeSH] 4. Social Participation [MeSH] 5. Rehabilitation, Vocational [MeSH] 6. Injustice OR justice OR social inclusion OR social exclusion OR social isolation OR social separation OR social barriers OR social distance OR social acceptance OR social	Table 1, page 7 Appendix 1

Reviewer	Original Manuscript and Reviewer Comments	Revised Manuscript and Author(s) Responses	Changes done on page/section
		rejection OR social participation OR deprivation OR marginalization OR alienation 7. 1 OR 2 OR 3 OR 4 OR 5 OR 6 (represents Social inclusion set) Set 2 1. Social stigma [MeSH] 2. Prejudice [MeSH] 3. Stigma OR prejudice OR stigmatise OR stigmatisation OR stigmatize OR stigmatization OR discrimination 4. 1 OR 2 OR 3 (represents Stigma set) Set 3 1. Occupational justice OR Occupational injustice OR Occupational deprivation OR Occupational alienation OR Occupational marginalisation OR Occupational imbalance OR Occupational OR occupation OR occupations OR activities OR work OR employment OR unemployment OR engagement (this set is used to narrow search to occupation as defined by OT not PubMed's definition) Set 4 1. Mental Disorders [MeSH] (this heading includes substance-related disorders) 2. Mentally Ill Persons [MeSH] 3. Mental disorders OR mental illness OR mentally ill OR Psychiatric disorder OR psychiatric illness OR psychological disorder OR Developmental Disability OR Intellectual Development Disorder OR Intellectual disability OR Mental retardation OR Mental deficiency 4. 1 OR 2 OR 3 (represents Mental Disorders set) Set 5	

Reviewer	Original Manuscript and Reviewer Comments	Revised Manuscript and Author(s) Responses	Changes done on page/section
		1. HIV [MeSH] 2. HIV Infections [MeSH] 3. Acquired Immunodeficiency Syndrome (MeSH) 4. HIV OR human immune deficiency virus OR AIDS OR acquired immunodeficiency syndrome OR acquired immune deficiency syndrome OR HIV/AIDS 5. 1 OR 2 OR 3 OR 4 Now combine Sets, 1 AND 2 AND 3 AND 4 or Sets 1 AND 2 AND 3 AND 5 Limit to last 20 years English Date: 31/01/19	
	The discussions of community participation, occupation rights and social justice at the best only mention mental illness and not HIV. The utility of discussing HIV and or mental illness remains unanswered and confusing, since it is not clear if the discussion is on people with disability, people living with HIV or people living with HIV who may have mental illness. The paper will be much stronger with sharper focus and connections between community participation, occupation rights and social justice and mental illness or HIV. If the paper discuss to discuss both mental health and HIV, it needs to state so and highlight what aspects of mental health and HIV it will focus on. This is an important topic and with revisions it is upto important information.	We thank the reviewer for their suggestions to make our discussion stronger. We have put this up front that although we were interested social inclusion and occupational justice in the duality of mental illness and HIV, to this end, the literature is scant and mainly focuses on mental illness and the concepts we explored. What our review suggests is that the limitations that you allude to in your comments are very present in the current research literature, and we have acknowledged it. This includes the fact that HIV and mental illness are not well documented as comorbid diagnoses in the literature on social inclusion and occupational justice. We believe that our study can be used as a foundation for future studies which would focus on specific elements of these concepts in targeted populations of people with mental illnesses and those living with HIV.	Discussion section. Page 26 Strengths and limitations. Page 3